# Numerical Modeling of the Natural and Manmade Factors Influencing Past and Current Changes in Polar, Mid-Latitude and Tropical Ozone

**Sergei P. Smyshlyaev [1,\*], Vener Y. Galin [2], Polina A. Blakitnaya [1] and Andrei R. Jakovlev [1]**

[1] Russian State Hydrometeorological University, Voronezhskaya Str., 79, 192007 Saint-Petersburg, Russia; pzim@rshu.ru (P.A.B.); endrusj@rambler.ru (A.R.J.)

[2] Institute of Numerical Mathematics RAS, Gubkina str., 8, 119991 Moscow, Russia, venergalin@yandex.ru

\* Correspondence: smyshl@rshu.ru

**Abstract:** A chemistry–climate model of the lower and middle atmosphere is used to compare the role of natural and anthropogenic factors in the observed variability of stratospheric ozone. Numerical experiments have been carried out on several scenarios of separate and combined effects of solar activity, stratospheric aerosol, sea surface temperature, greenhouse gases, and ozone-depleting substances emissions on ozone for the period from 1979 to 2020. Simulations for the past and present periods are compared to the results of ground-based and satellite observations. Estimates of observed trends in column total ozone for the entire period 1980–2018 and separately for the late twentieth and early twenty-first century are presented.

**Keywords:** stratospheric ozone; natural and anthropogenic factors; numerical modeling; satellite observations; trend estimations

## 1. Introduction

The change in the content of atmospheric ozone, which protects life on Earth from the harmful effects of the hard part of the ultraviolet radiation of the Sun, has attracted the attention of scientists and policymakers for many years in connection with the observed negative trends in ozone global content since the early 1980s [1–5]. In this regard, attention has concentrated on the impact of anthropogenic factors associated with industrial and household emissions of chlorine and bromine gases (ozone-depleting species—ODS) into the atmosphere, contributing to the destruction of the ozone layer. ODS are inert in the lower atmosphere, but, while rising into the stratosphere, they fall under the influence of intense fluxes of ultraviolet (UV) solar radiation. Thus, they are destroyed and release a large amount of chlorine and bromine radicals that destroy ozone molecules in catalytic cycles [6,7]. The rapid and timely adoption of measures to limit ODS emissions under the Montreal Protocol and its subsequent amendments has halted the growth of ODS in the stratosphere, and the stratospheric ozone is now expected to begin to recover to early 1980s levels [8,9].

Studies of trends in total ozone in the late twentieth and early twenty-first centuries depicted that they were negative until the mid-90s and in subsequent years showed signs of a weak, mostly insignificant positive trend [1–5,8,9]. At the same time, significant trends were observed in the upper and lower stratosphere, whereas no changes were observed in the middle stratosphere, where the largest amount of ozone is located [10]. At the same time, trends in different latitudinal regions differ quantitatively [11,12]. The magnitude, significance, and even sign of the trend over the last period can change significantly with each additional year of observation, so despite previously published results of ozone trend estimates, new results that take into account recent measurements may be useful to the scientific community.

Meanwhile, the results of measurements show that, despite the measures taken to limit the impact of anthropogenic factors on ozone, its recovery has not been as fast as expected [11–14], and in some regions in the lower stratosphere there still is a slight decrease [15]. These facts make timely and relevant the detailed study of the influence of natural factors such as solar activity, stratospheric aerosol content, temperature of the atmosphere and ocean, the area covered by ocean ice, and the general circulation of the atmosphere on the content of stratospheric ozone. The combined effects of natural and anthropogenic factors can lead to non-linear effects in ozone variability, weighed down, in addition, by a large number of positive and negative feedbacks superimposed on the previously expected faster recovery of the ozone layer as a result of a reduction in ODS [16,17].

To study all the features of nonlinear interactions between physical and chemical processes in different altitude layers of the atmosphere that affect the change of stratospheric ozone content under the influence of natural and anthropogenic factors, the most rational approach utilizes numerical modeling of chemical and climatic processes in the atmosphere, taking into account their interaction and feedbacks, and comparing the results of numerical modeling to the results of observations [18,19]. This approach is proposed in this paper, combining the analysis of satellite observations of total ozone from the early 80s to the present time with numerical modeling of atmospheric ozone variability from 1980 to 2020, taking into account the influence of chemical and dynamical factors of ozone variability as well as their interaction.

The purpose of this work is to study the trends of total ozone in different latitudinal regions during different periods of time from the early 80s of the twentieth century to the end of the 10s of the twenty-first century to study the influence of natural, anthropogenic, physical, and chemical factors on the observed variability of total ozone in different latitudinal regions and to highlight the predominant role of individual factors in individual time periods.

## 2. Methodology

To study the interannual variability of the total column ozone in the tropical, middle, and polar latitudes during 1980–2020, the results of satellite observations and numerical modeling using a chemistry–climate model (CCM) were analyzed. Satellite observations of solar backscatter ultraviolet (SBUV), total ozone mapping spectrometer (TOMS), and ozone monitoring instrument (OMI) as well as the results of calculations using a CCM of the Institute of Numerical Mathematics and the Russian State Hydrometeorological University (INM RAS–RSHU CCM), were averaged for each year and the latitude bands 90S–60S (Antarctica), 60S–30S (middle latitude southern hemisphere), 30S–30N (tropics), 30N–60N (middle latitude northern hemisphere), and 60N–90N (the Arctic).

The solar backscattering ultraviolet (SBUV) radiometer [20] is a series of operational remote sensors on Nimbus and NOAA weather satellites in solar synchronous orbits that provide measurements of total ozone with global coverage in the stratosphere, as well as ozone profiles, from 1970 to 2017 [21]. Data from different satellites are calibrated among themselves, resulting in a merged set of combined data [9], continuously covering the period from 1979 to 2017.

The Total Ozone Mapping Spectrometer (TOMS) is a NASA satellite instrument for measuring ozone concentration [22–25]. Of the five instruments built, four entered successful orbit on the Nimbus-7, Meteor–3, Advanced Earth Observing Satellite (ADEOS), and Earth Probe (EP) satellites. Instruments on the Nimbus-7 and Meteor–3 satellites produced a continuous set of data on the total ozone content from November 1978 to December 1994 [23, 24]. The instrument on the ADEOS satellite gave data from 1996 to mid-1997, but did not cover a full year. Since mid-1996, when the Earth Probe satellite was launched [25], the TOMS instrument on that satellite transmitted data on the total ozone content until 2006. As a continuation and replacement of the TOMS series of instruments, an ozone monitoring instrument (OMI) was launched on the Aura satellite in 2004, which provides data on the total ozone content to date [26].

Due to the fact that only SBUV satellite measurements provide consistent data on the total ozone content for the entire study period from 1980 to 2018, they were used in this work to study trends in ozone content, both for the entire period and for individual time intervals. TOMS and OMI data that do not allow for a consistent series for the entire study period were considered complementary to the

SBUV data to clarify trends in individual time periods. For the average annual SBUV data values in each latitudinal region, linear trends of total ozone were calculated using linear regression formulas [27] for the entire analyzed period from 1980 to 2020 and separately for the end of the twentieth (1980–2000) and the beginning of the twenty-first (2000–2017) centuries. As a quantitative characteristic of the trend, the linear regression coefficient characterizing the rate of change in the total ozone content (Dobson units per year) was calculated. For the end of the twentieth century, when there was a sharp transition on a global scale from a stable decrease in ozone from 1980 to the mid-90s to a sharp increase from the mid-90s to the early 2000s, separate trends were estimated for these two multidirectional periods of variability in total ozone for different latitudinal regions. For all calculated trends, their statistical significance was estimated by Fisher's criterion [27] by comparing the coefficient of the linear trend with the spread of data relative to the regression line, and then the probability density function (PDF) and root mean square (RMS) error were evaluated.

To assess the role of various factors affecting the interannual variability of ozone content, numerical experiments were performed with the INM RAS–RSHU CCM (version 1) [28]. The model has a spatial resolution of 4 × 5 degrees latitude/longitude and covers the altitude range from the Earth's surface up to the mesopause (around 90 km) with 39 sigma-pressure levels with variable pressure spacing [28]. In order to take into account an interaction between chemical and dynamical processes in the low and middle atmosphere under the frame of a CCM, a General Circulation Model (GCM), from the Institute of Numerical Mathematics of the Russian Academy of Sciences (INM RAS), is interactively coupled with a global chemistry-transport model (CTM) of the Russian State Hydrometeorological University (RSHU), which was designed based on the three-dimensional extension of the SUNY-SPB two-dimensional model [29–32]. The CCM calculates the variability of 74 chemically active gases in the lower and middle atmospheres, which are interacting in 174 gas-phase and heterogeneous chemical reactions and 51 photodissociation processes, including oxygen, nitrogen, hydrogen, chlorine, bromine, carbon, and sulfur cycles [33,34]. The model considers the processes of the formation and evolution of polar stratospheric clouds (PSC) based on stratospheric sulfate aerosol [35–37].

The influence of ozone-depleting substances on atmospheric chemistry is taken into account in the model by setting the concentrations of freons and halons at the lower boundary with subsequent calculation of their chemical evolution and photodissociation in the troposphere and stratosphere, taking into account their transport by advection and turbulent fluxes. The influence of spectral solar radiation fluxes is taken into account by specifying their variability at the upper boundary of the model (about 90 km), taking into account the attenuation of the overlying layers of the atmosphere and calculating their attenuation due to absorption and scattering by gases and aerosols. The total flux of direct and scattered radiation is further used to calculate both the rates of photodissociation of atmospheric gases and the heating of the atmosphere by short-wave radiation. The long-wave outgoing radiation of the earth's surface and atmosphere is calculated to estimate the cooling of the atmosphere. The aerosol influence was taken into account by specifying the aerosol surface area in a unit volume, which was used to calculate the rates of heterogeneous reactions, to estimate the optical thickness of the aerosol attenuation of solar radiation and as a basis for the formation of polar stratospheric clouds at low temperature. The sea surface temperature and the sea ice coverage are taken into account when setting the lower boundary condition for the thermodynamics equation in GCM, which affects the temperature and, accordingly, the rate of chemical reactions, as well as the wind speed and the transport of ozone and related gases.

To assess the role of the main factors influencing the interannual variability of ozone content, emissions of chlorine and bromine substances contributing to ozone depletion (ODS), set on the basis of the World Meteorological Organization (WMO) scenarios of 2011 [2]; changes in solar radiation fluxes at the upper atmosphere set according to the empirical Naval Research Laboratory (NRL) model [38]; stratospheric sulfate aerosol content, set according to [39], and sea surface temperature (SST) set according to the Met-Office re-analysis [40], in addition to the baseline scenario, which takes into account the influence of all these factors on the interannual variability of ozone content (basic scenario). Additional model numerical experiments were carried out, which took into account the

influence of only each of these factors separately. A comparison of the results of model experiments under these scenarios allowed us to estimate the role of each of the tested factors in the interannual variability of the total ozone content in different latitudinal zones. The interannual variability of the studied factors influencing ozone trends is shown in Figure1.

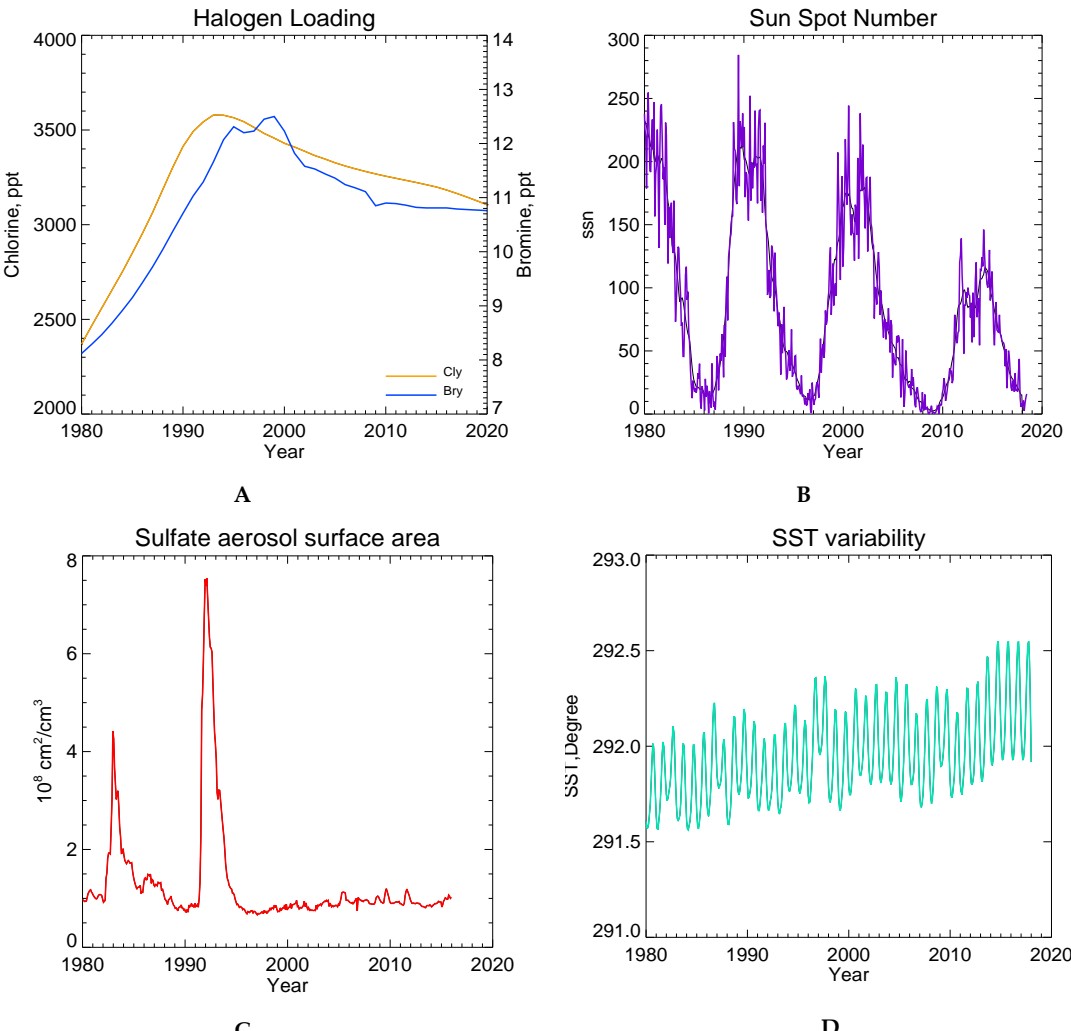

**Figure 1.** Interannual variability of the tested factors influencing ozone trends: (**A**) chlorine and bromine ozone destroyers; (**B**) sunspot number (SSN) variability [41]; (**C**) stratospheric aerosol; (**D**) sea surface temperature.

## 3. Results

### 3.1. Total Column Ozone Interannual Variability in the Tropics

Both satellite measurements and numerical simulations allow us to select three periods of interannual variability in total ozone in the tropics (Figure 2): a significant drop from the early 80s to the mid-90s of the twentieth century, a sharp increase in ozone from the mid-90s to the early 2000s, and weak variability during the first 20 years of the twenty-first century. This variability is superimposed by short-term fluctuations in total ozone with highs around 1990, 2000, and 2010, i.e., almost every 10 years, and lows around 1986–1987, 1993–1994, 2005–2007, and 2016. Trend estimates from SBUV satellite measurements for the entire study period (1980–2018) and separately in the twentieth and twenty-first centuries show a general trend of a weak decrease in total ozone over all three periods. At the same time, according to Fisher's criterion, these trends have a significance with

a probability density function (PDF) of 92% for the general trend and the trend of the late twentieth century and 0.59% for the trend of the twenty-first century. The coefficients of the linear trend are as follows: period 1980–2018 −0.06 Dobson units (DU) per year, period 1980–2000 −0.17 DU per year, period 2000–2018 −0.07 DU per year. Thus, the trends for the entire study period and the twentieth century are significant with a 10% level of significance, and the trend of the twenty-first century can be considered insignificant. If we use our trend estimates from the three aforementioned periods, we find that the linear trend coefficient for 1980–1995 is −0.35 DU per year with PDF = 0.98 (i.e., a 5% significance level), and the coefficient for 1995–2000, when there was a sharp increase in the total ozone content, was 0.87 DU per year.

Analysis of the results of model experiments on the sensitivity of the tropical ozone to the variability of influencing factors (Figure 2B) shows that the basic tendency of ozone variability during the entire tested period is influenced by the increase in chlorine and bromine gases during the late twentieth century as well as their reduction in the early twenty-first century, determined on the basis of WMO 2011 scenarios. Comparisons with satellite measurements shown in Figure 2A reveal that WMO 2011 scenarios are likely overestimating the reduction of chlorine and bromine gases in the twenty-first century, which is reflected in a significant excess of the model results from satellite observations at the end of the tested period, namely in 2010–2020. At the same time, at the beginning of the period, i.e., 1980–1985, the importance of chlorine and bromine ozone depletion is probably also underestimated in WMO 2011 scenarios, which is also evidenced by a significant excess of model values from satellite observations. The period of rapid reduction of column ozone in the first half of the 90s, indicated both by SBUV observations and model calculations (Figure 2A), is associated not only with increased halogen loading with maximum effect on ozone at the end of the 90s (Figure 2B) but also with the influence of other factors, in particular, the increase in the content of sulfate stratospheric aerosol as a result of emissions by mount Pinatubo (peak aerosol effect on ozone at 1992) (Figure 2B), solar activity (ozone minimum at 1995–1996, Figure 2B), and sea surface temperature (peak influence on ozone at 1995). At the end of the 90s and beginning of the 2000s, when halogen loading was at a maximum (Figure 2B—orange line), both satellite data and model calculations revealed the increase in column ozone (Figure 2A).

Comparison of the Figure 1B with Figure 2A,B allows us to conclude that solar activity, particularly its 11-year cycle, is responsible for the observed positive peaks of tropical ozone in 1990, 2002–2003, and 2014–2015 as well as for the low column ozone in 1986–1987 and 2007–2008. The change in stratospheric sulfate aerosol loading (Figure 1C) plays an important role in ozone depletion not only after the Mt. Pinatubo emissions in the first part of the 90s but also after the emissions of the El Chichon in the mid-80s (Figure 2B). The sea surface temperature's impact on the column ozone is visible at the end of the 90s after the major El Nino southern oscillation (ENSO) event in 1997–1998 resulted in the enhanced SST (Figure 1D) and column ozone increase in the tropics (Figure 2B—green line). The changes caused by ENSO in the dynamics of ozone and related gas transport contribute to a sharp increase of tropical ozone in the late 90s of the twentieth century, along with solar activity and purification of the stratosphere from the volcanic aerosol. Overall, it should be noted that the results of the numerical experiments demonstrated that the changes in chlorine and bromine loading regulated by the Montreal Protocol and its amendments are responsible for long-term decadal variability in the column ozone, while its short-term changes with time scale around several years are mostly defined by natural factors, such as the 11-year solar cycle, stratosphere sulfate aerosol loading, and variability of the dynamic processes initiated by sea surface temperature changes. The simultaneous influence of these factors on the decreasing in the column ozone in the early 90s and increasing the column ozone in the latter part of the 90s led to drastic, sharp changes in the content of tropical ozone. At the same time, the influence of the anthropogenic chlorine–bromine factors in these processes is not dominant.

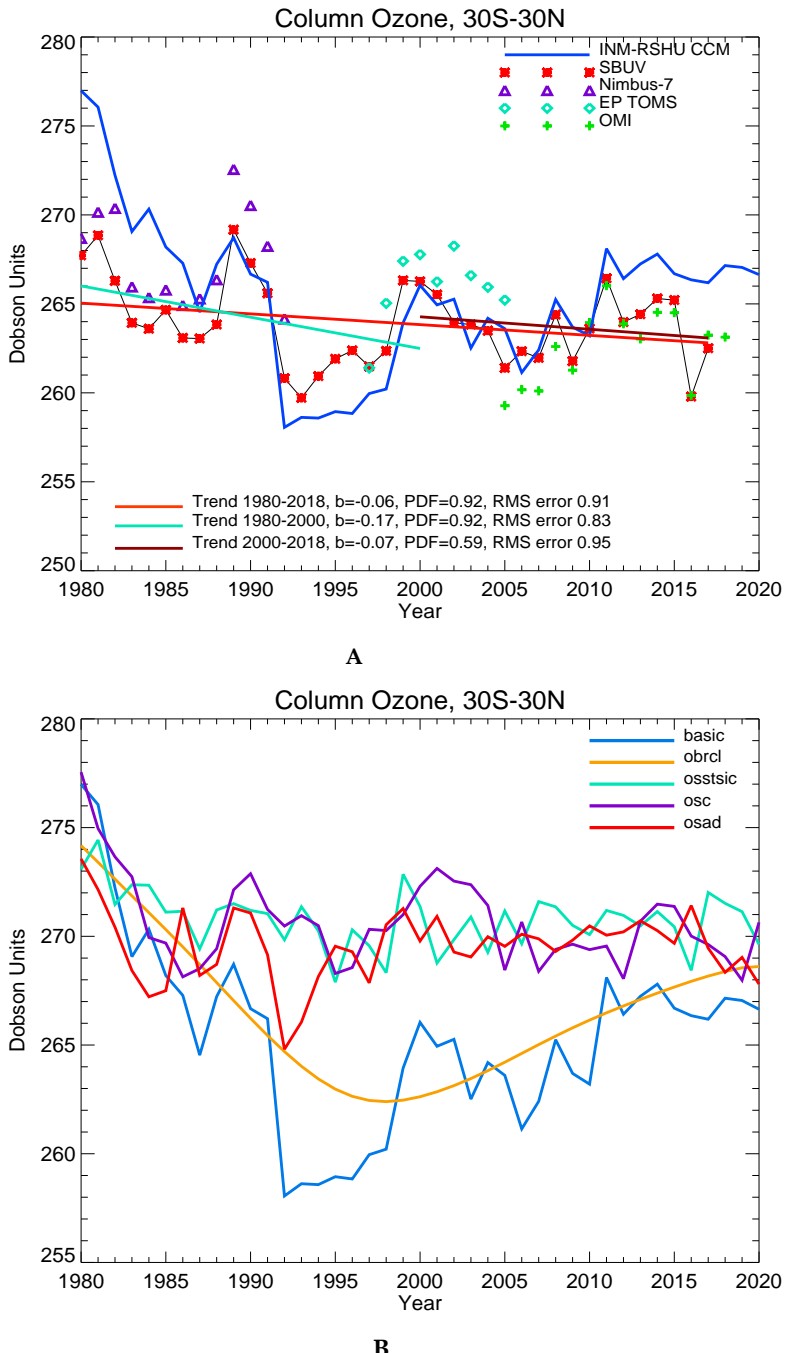

**Figure 2.** Interannual variability of column ozone in the tropics: (**A**) Results of satellite measurements (solar backscattering ultraviolet (SBUV), Nimbus-7 Total Ozone Mapping Spectrometer (TOMS), Earth Probe (EP) TOMS, and the Ozone Monitoring Instrument (OMI)) and numerical modeling (chemistry–climate model of the Institute of Numerical Mathematics and the Russian State Hydrometeorological University) with an assessment of trends (b—coefficient of linear regression) and their significance (PDF—probability density function, and RMS—root mean square error); (**B**) results of numerical modeling under scenarios with a separate action of influencing factors: basic— all factors' interannual variabilities are taken into account, obrcl—only the surface emissions of bromine and chlorine-containing gases' interannual variability is taken into account, osstsic—only sea surface temperature and sea ice coverage interannual variabilities are taken into account, osc— only solar radiation's interannual variability is taken into account, osad—only sulfate aerosol density's interannual variability is taken into account.

### 3.2. Total Column Ozone Interannual Variability in the Mid-Latitudes of the Northern Hemisphere

For the middle latitudes of the northern hemisphere (Figure 3), satellite observations indicate the same three periods of steady change in total column ozone as for tropical latitudes. Of particular note is a significant decrease in ozone from the early 1980s to the mid-1990s, a sharp increase in its content in the second half of the 1990s, and a weak declining in the column ozone in the twenty-first century. Quantitative trend estimates show a greater rate of decrease in the northern mid-latitude ozone compared to that of tropical latitudes. In particular, for the entire considered period of 1980–2020, the linear trend coefficient is −0.19 Dobson Units (DU) per year with PDF = 0.97. For the end of the twentieth century, the linear trend coefficient is −0.76 DU per year with PDF = −0.99, and for the beginning of the twenty-first century, the linear trend coefficient is −0.11 DU per year with PDF = 0.43. Thus, it can be stated that negative trends in the northern mid-latitude column ozone are significant with a level of 5% for the entire period of 1980–2020 and for the end of the twentieth century (1980–2000). For the beginning of the twenty-first century, the trend can be considered insignificant. Column ozone trend estimation for the three earlier allocated periods indicate that in 1980–1995, the coefficient of the linear trend is −1.41 DU per year with PDF = 0.99, and in 1995–2000, the coefficient of the positive linear trend is equal to 1.42 DU per year.

The analysis of factors affecting northern mid-latitude column ozone demonstrates (Figure 3B) that chlorine and bromine loading in accordance with WMO scenarios leads to significant column ozone reduction in the northern mid-latitudes during the first half of the research period (1980–1995) and to column ozone rise at the end of the research period (2010–2020). From the mid-90s to 2010, the influence of halogen gases on the northern mid-latitude column ozone is negligible. Other factors include the greater influence of stratospheric aerosol at the end of the twentieth century compared to tropical latitudes, especially during periods of volcanic eruptions, and the greater role of sea surface temperature variability comparable to solar activity. In periods of sharp decrease and then ozone recovery in the 90s similar to tropical latitudes, a combination of several factors played an additive role: the fall in ozone as a result of increased chlorine and bromine gas content, the role of sea surface temperature on the background of increasing chlorine and bromine gases, the increase in the ozone content as a result of the purification of stratospheric aerosol, the influence of the variability of sea surface temperature, and the rise and reduction of solar activity.

### 3.3. Total Column Ozone Interannual Variability in the Mid-Latitudes of the Southern Hemisphere

The column ozone variability in the southern middle latitudes (Figure 4) is, on the one hand, characterized by the presence of a fairly pronounced negative trend throughout the period of 1980–2020 with a higher coefficient of −0.33 DU per year and PDF = 0.99. On the other hand, the periods of sharp decrease and then increase in ozone content in the 90s, clearly distinguished in the tropics and mid-latitudes of the northern hemisphere and in the Southern hemisphere, have a different look according to the SBUV (Figure 4A). In particular, there are two column ozone minima in 1993 and 1997, two maxima in 1996 and 1998, and growth in the late twentieth century that continues until 2002. As a result, if, as was done for the tropical and middle latitudes of the Northern hemisphere, we were to evaluate trends from 1980 to 1995 and then from 1995 to 2000, the trend in 1980–1995 differs little from the trend for the overall period of 1980–2000 (coefficient −1.05 DU per year vs. −0.9 DU per year), and there is an insignificant column ozone trend (PDF = 0.58) from 1995 to 2000.

Thus, for the middle latitudes of the southern hemisphere, it is possible to propose a different allocation of periods of steady change in ozone content compared to the other latitudes: the period of decrease in ozone content is extended from 1980 to 1997, and the period of growth covers the years from 1997 to 2002, and third period from 2002 to 2018. Then, for these periods, the linear trend coefficients are −1.0, 2.0, and −0.01 DU per year, respectively, and the values of PDF are 0.99, 0.97, and 0.05, respectively. In addition, for the twenty-first century in the mid-latitudes of the Southern hemisphere, a quasi-two-year cycle of changes in the total ozone content is noticeable.

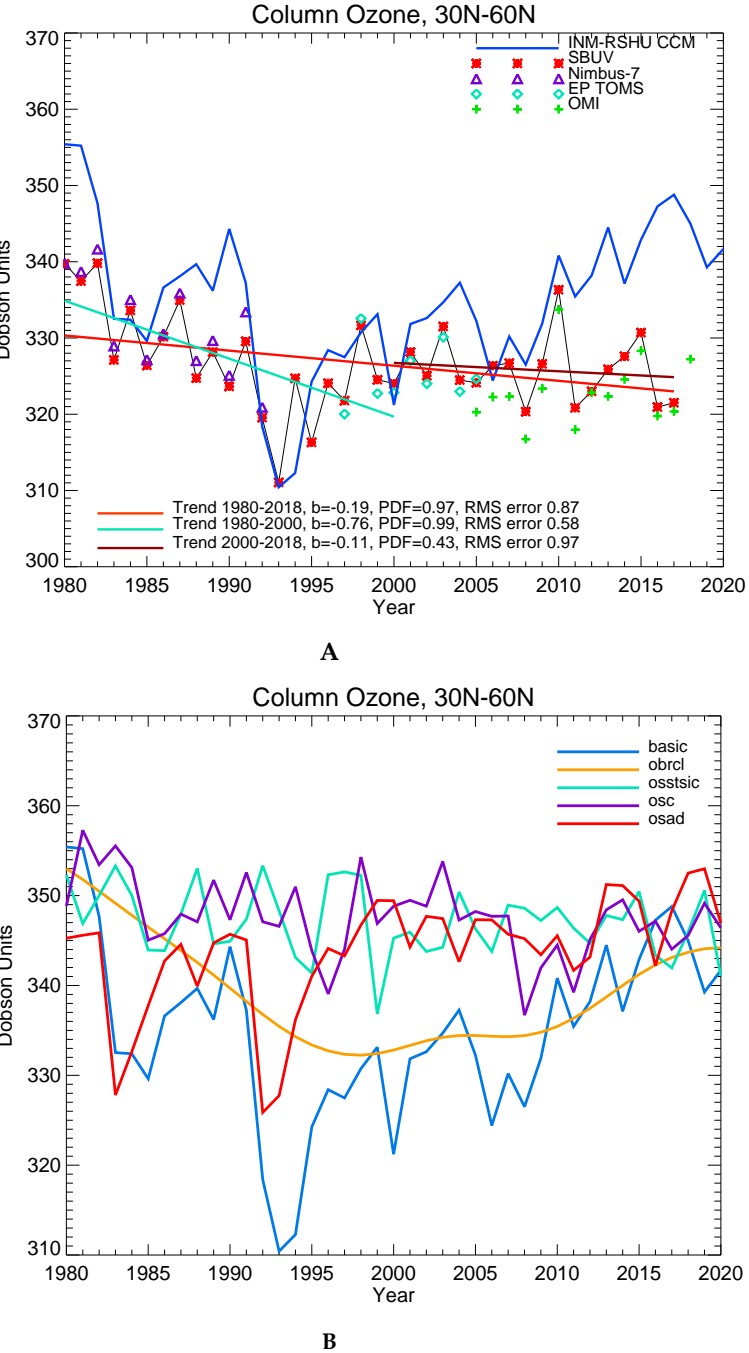

**Figure 3.** Interannual variability of column ozone in the northern mid-latitudes: (**A**) Results of satellite measurements (solar backscattering ultraviolet (SBUV), Nimbus-7 Total Ozone Mapping Spectrometer (TOMS), Earth Probe (EP) TOMS, and the Ozone Monitoring Instrument (OMI)) and numerical modeling (chemistry—climate model of the Institute of Numerical Mathematics and the Russian State Hydrometeorological University) with an assessment of trends (b—coefficient of linear regression) and their significance (PDF—probability density function, and RMS—root mean square error); (**B**) results of numerical modeling under scenarios with a separate action of influencing factors: basic—all factors' interannual variabilities are taken into account, obrcl—only the surface emissions of bromine and chlorine-containing gases' interannual variability is taken into account, osstsic—only sea surface temperature and sea ice coverage interannual variabilities are taken into account, osc—only solar radiation's interannual variability is taken into account, osad—only sulfate aerosol density's interannual variability is taken into account.

Analysis of influencing factors ((Figure 4B) reveals that stratospheric sulfate aerosol in the mid-latitudes of the southern hemisphere plays a smaller role than noted both in the tropics and in the northern hemisphere, with the model likely overestimating the importance of this factor in the southern hemisphere. The impact of chlorine and bromine loading according to WMO 2011 scenarios on changes in the southern hemisphere's mid-latitude ozone is also overestimated by the model, especially at the end of the period (2010–2020). Interestingly, the influence of solar activity is manifested differently in the mid-latitudes of the northern and southern hemispheres, despite the fact that the variability of the spectral fluxes of solar radiation is the same for the entire globe, according to NRL data. In particular, in a scenario that takes into account only the variability of solar flows (Figure 4B), there are significant ozone peaks in 1991, 1997, 2002, and 2005, which are also evident in satellite observations. These peaks are not obviously related to the 11-year solar cycle, but more to the short-term variability of the solar fluxes (Figure 1B). In the model scenario, taking into account the influence of all factors, some of these maxima are offset by the influence of other factors, in particular the overestimated role of stratospheric sulfate aerosol. The effect of sea surface temperature variability is comparable in order of magnitude to that of solar activity and, in some years, enhances it (e.g., 2002) while in some years it compensates for it (e.g., 2000). Overall, the model's overestimation of the chemical factors of ozone depletion in the southern mid-latitudes is, probably, a result of underestimating the southern dynamical processes' roles in the model as well as overestimating the Antarctic ozone chemical depletion, visible in Figure 6.

*3.4. Total Column Ozone Interannual Variability in the Polar Regions*

In the Arctic zone (Figure 5), the general trend of ozone content change is much smaller than observed at other latitudes (the linear trend coefficient for 1980–2020 is −0.22 DU per year) with relatively low significance (PDF = 0.83), whereas the twentieth century trend has a coefficient of −1.21 DU per year and PDF = 0.99, and the twenty-first century's trend is negative and insignificant (PDF = 0.18).

The analysis of influencing factors (Figure 5B) shows that, first, as in other regions in the Arctic, the role of chlorine and bromine gases is overestimated in 2010–2020, resulting in a significant excess of simulation results over satellite data. Secondly, the important role of dynamic factors in the model experiments reflect the influence of sea surface temperature on the column ozone, probably due to the impact of dynamical processes on the stability of the polar vortex and heat and mass exchange between the Arctic and mid-latitudes. Thirdly, there is a combined influence of solar activity and stratospheric aerosol, which in some years gives the strengthened maxima coinciding with phases of mid-latitudes, but is not visible in the results of satellite observations in the Arctic (as for example in 1990). This indicates the underestimated role of dynamic factors in the model.

In Antarctica (Figure 6), on the one hand, there are two periods: the first with a significant negative trend of column ozone from 1980 to 2000 (linear trend coefficient −2.26 DU per year and PDF = 0.99), and, on the other hand, an insignificant positive trend in the twenty-first century (PDF = 0.56). The results of the simulation, in contrast to most other regions in the Antarctic, are substantially underestimating the ozone content.

While assessing the influencing factors (Figure 6B), first of all, it should be noted that, unlike other regions, the impact of chlorine and bromine loading does not lead to significant column ozone rise at the end of the considered period (2010–2020), whereas at the beginning of this period (1980–1995), the influence of halogen gases is maximum compared to other latitudes (compare the orange lines in Figures 2B, 3B, 4B, 5B and 6B). Thus, the weak sign of Antarctic ozone recovery in the twenty-first century may not merely be related to measures directed to reducing the emissions of ozone-depleting substances.

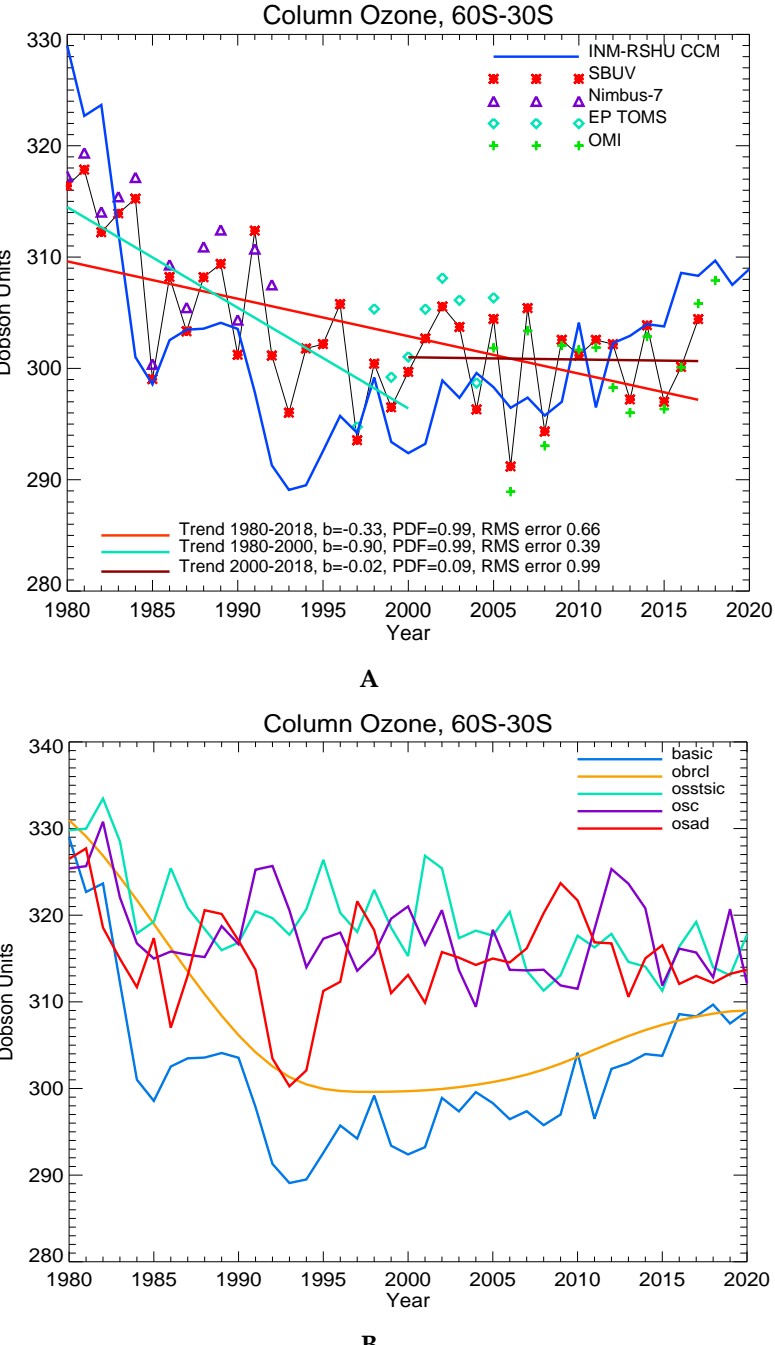

**Figure 4.** Interannual variability of column ozone in the southern mid-latitudes: (**A**) Results of satellite measurements (solar backscattering ultraviolet (SBUV), Nimbus-7 Total Ozone Mapping Spectrometer (TOMS), Earth Probe (EP) TOMS, and the Ozone Monitoring Instrument (OMI)) and numerical modeling (chemistry—climate model of the Institute of Numerical Mathematics and the Russian State Hydrometeorological University) with an assessment of trends (b—coefficient of linear regression) and their significance (PDF—probability density function, and RMS—root mean square error); (**B**) results of numerical modeling under scenarios with a separate action of influencing factors: basic—all factors' interannual variabilities are taken into account, obrcl—only the surface emissions of bromine and chlorine-containing gases' interannual variability is taken into account, osstsic—only sea surface temperature and sea ice coverage interannual variabilities are taken into account, osc—only solar radiation's interannual variability is taken into account, osad—only sulfate aerosol density's interannual variability is taken into account.

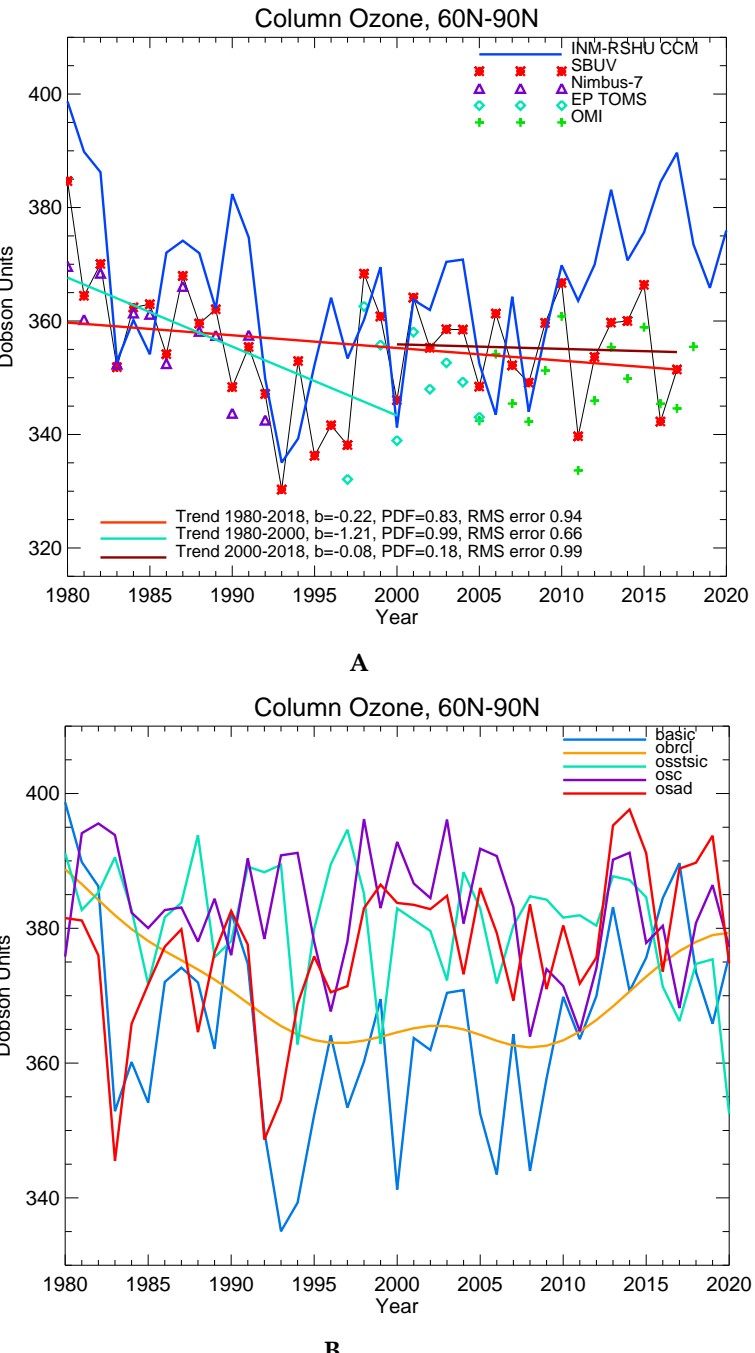

**Figure 5.** Interannual variability of column ozone in the northern polar latitudes: (**A**) Results of satellite measurements (solar backscattering ultraviolet (SBUV), Nimbus-7 Total Ozone Mapping Spectrometer (TOMS), Earth Probe (EP) TOMS, and the Ozone Monitoring Instrument (OMI)) and numerical modeling (chemistry–climate model of the Institute of Numerical Mathematics and the Russian State Hydrometeorological University) with an assessment of trends (b—coefficient of linear regression) and their significance (PDF—probability density function, and RMS—root mean square error); (**B**) results of numerical modeling under scenarios with a separate action of influencing factors: basic—all factors' interannual variabilities are taken into account, obrcl—only the surface emissions of bromine and chlorine-containing gases' interannual variability is taken into account, osstsic—only sea surface temperature and sea ice coverage interannual variabilities are taken into account, osc—only solar radiation's interannual variability is taken into account, osad—only sulfate aerosol density's interannual variability is taken into account.

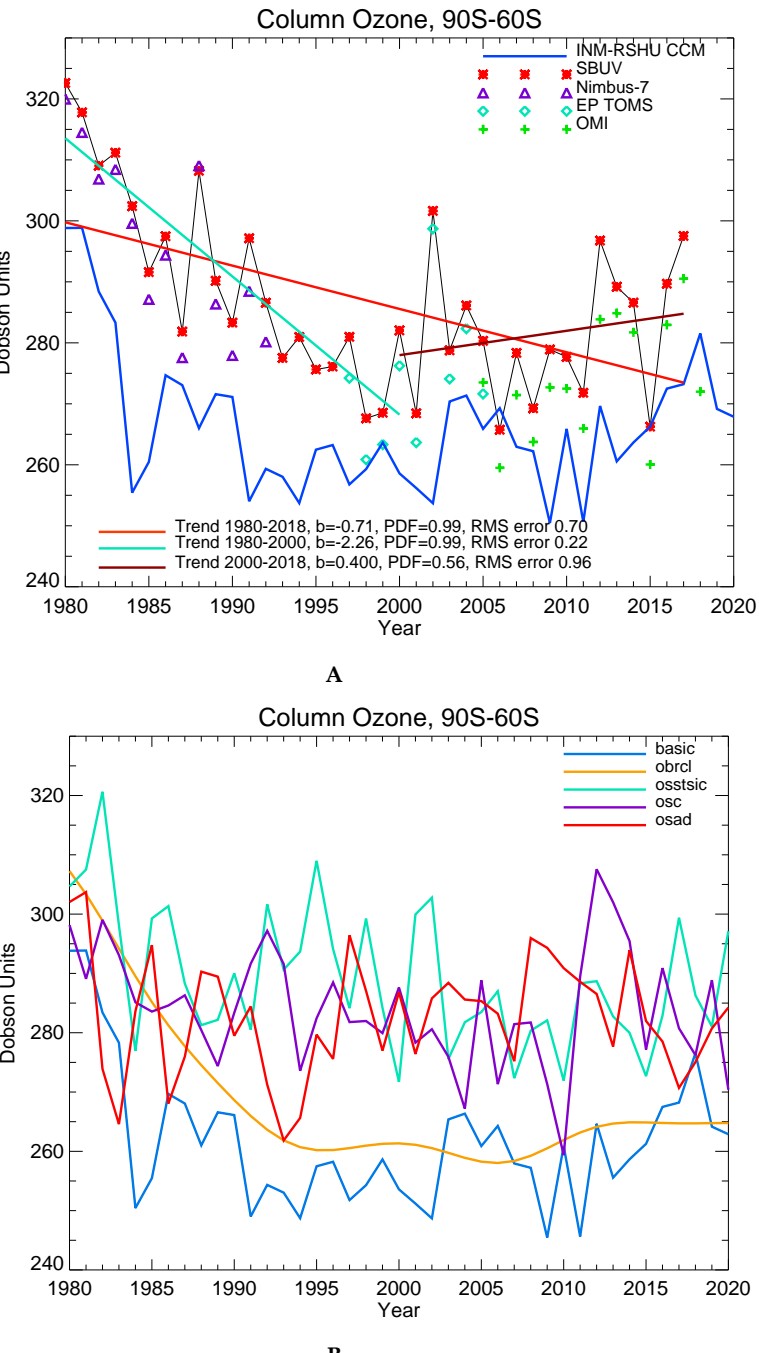

**Figure 6.** Interannual variability of column ozone in the southern polar latitudes: (**A**) Results of satellite measurements (solar backscattering ultraviolet (SBUV), Nimbus-7 Total Ozone Mapping Spectrometer (TOMS), Earth Probe (EP) TOMS, and the Ozone Monitoring Instrument (OMI)) and numerical modeling (chemistry–climate model of the Institute of Numerical Mathematics and the Russian State Hydrometeorological University) with an assessment of trends (b—coefficient of linear regression) and their significance (PDF—probability density function, and RMS—root mean square error); (**B**) results of numerical modeling under scenarios with a separate action of influencing factors: basic—all factors' interannual variabilities are taken into account, obrcl—only the surface emissions of bromine and chlorine-containing gases' interannual variability is taken into account, osstsic—only sea surface temperature and sea ice coverage interannual variabilities are taken into account, osc—only solar radiation's interannual variability is taken into account, osad—only sulfate aerosol density's interannual variability is taken into account.

## 4. Conclusions and Discussion

In this paper, the study of the column ozone in the tropical, mid, and polar latitudes is presented for the period from 1980 to 2018. Analyses of the results of satellite observations demonstrated that for all considered latitudinal regions, there is a significant negative trend in the annual average column ozone during the entire study period. At the same time, the level of significance for the tropics is 10%, for the middle latitudes of the northern and southern hemispheres and Antarctica is 5%, and for the Arctic zone it is 20%. Thus, the least significant negative trend is observed in the northern polar zone, and the most significant negative trend of total ozone (PDF = 0.99) is in the middle and polar latitudes of the southern hemisphere.

When considering individual trends for the variability of total ozone in the twentieth and twenty-first centuries of the considered time period for all latitudinal zones, a significant negative trend is found at the end of the twentieth century (1980–2000) and an insignificant trend is at the beginning of the twenty-first century (2000–2018). At the same time, a positive albeit insignificant trend is observed only in Antarctica in the twenty-first century. The significance level for all latitudinal zones except tropical for the late twentieth century is 1%, while in the tropical zone it is 10%. A more detailed examination of interannual variability allows us to distinguish in all three latitudinal bands a period of steady variability in the column ozone: a strong decrease since the beginning of the 80s until the mid-90s, the rapid growth since the mid-90s to early twenty-first century, and then an insignificant column ozone trend in the twenty-first century. At the same time, while in the northern hemisphere and the tropics the duration of the first period of rapid ozone depletion is 13–15 years (until 1993–1995), in the southern hemisphere this period lasts until 1997–1998, i.e., 3–4 years longer. In addition, the period of rapid growth in the late 90s to the early 2000s is also different for the two hemispheres. In the northern hemisphere, it ends almost at the turn of the century, and it lasts two years longer in the southern hemisphere (until 2002).

An analysis of the significance of individual factors, based on the results of numerical experiments with the chemical-climate model, showed that WMO 2011 scenarios probably overestimating the reduction of ozone-depleting gas emissions into the atmosphere in 2010–2020. As a result, model calculations for all latitudinal zones depict a fairly rapid increase in column ozone, while satellite observations do not confirm this. In the tropics, the rise in column ozone due to reduction in the halogen loading from the late 90s till 2020 is estimated as uniform with a total of 7 Dobson units. In contrast, in the middle latitudes of the northern hemisphere until 2010, the model reveals a weak increase of 2 Dobson units and then a stronger increase of 10 Dobson units from 2010 to 2020. In the mid-latitudes of the southern hemisphere, the halogen factor has little effect on ozone from 1995 to 2006, and then leads to an increase in ozone at 9 Dobson units by 2020. In both polar regions from 1995 to 2010, the ozone content changes very little despite changes in emissions of chlorine and bromine gases. Finally, there is a 16 Dobson-unit rise in column ozone in the Arctic, and a 3 Dobson-unit rise in Antarctica.

Other factors include the predominance of solar radiation flux variability in the tropics and the mid-latitudes of the southern hemisphere, stratospheric aerosol loading in the mid-latitudes of the northern hemisphere, and sea surface temperature in the polar regions. The influence of quasi-biennial oscillation in the mid-latitudes of the southern hemisphere in the twenty-first century should also be noted, and are quite clearly presented in the results of satellite observations and only slightly noticeable in the results of modeling.

**Author Contributions:** Conceptualization, S.P.S., V.Y.G.; Methodology, S.P.S., V.Y.G.; Formal analysis, P.A.B.; Data curation, A.R.J.; Writing—original draft preparation, S.P.S.; Writing—review and editing, S.P.S.; Visualization, A.R.J. All authors have read and agreed to the published version of the manuscript.

**Funding:** This research was funded by the Russian Science Foundation, grant: 19-17-00198. Polar processes were studied under the Russian Foundation for Basic Research, grant: 17-05-01277.

**Acknowledgments:** We would like to thank National Aeronautics and Space Administration's Goddard Space Flight Center for providing the SBUV Merged Ozone Data Set, and Atmospheric Chemistry and Dynamics Laboratory (Code 614) for providing us with the Nimbus-7 and Earth Probe TOMS data as well as the Aura OMI instrument column total ozone data.

**Conflicts of Interest:** The authors declare no conflict of interest.

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
