# Peer review of "Numerical Modeling of the Natural and Manmade Factors Influencing Past and Current Changes in Polar, Mid-Latitude and Tropical Ozone"

_atmosphere, doi:10.3390/atmos11010076_

Round 1

Reviewer 1 Report

The study compares some characteristics of temporal evolution of total ozone derived from results of SBUV measurements and numerical modeling. Model experiments follow different scenarios that take or do not take into account some prescribed factors that can have direct or indirect effects on stratospheric ozone. These factors are halogen loading, stratospheric aerosol loading, solar activity, and sea surface temperature. Comparing the model results to the observed total ozone variability the authors deduce about potential role of the factors in total ozone change. An important result of the work is that the change in chlorine and bromine loading regulated by the Montreal protocol and its amendments is, probably, only in part responsible for the observed changes in total ozone for the period since 1979.

The subject of the manuscript is relevant to the journal. However some analysis carried out and findings are not fully clear and/or justified to me. I detailed below my comments, suggestions and questions to the authors.

General comments

The manuscript contains a lot of grammatical and syntactic mistakes. Some words are of inadequate meaning. These complicate understanding.

The title is not accurate enough. The latest year is 2020, and there is no reason to declare future changes. Further, “anthropogenic factors” in the title sound vaguely.

The review of previous works on the subject is very poor. The scientific novelty of the paper is not specified.

The course of thought of the authors is not always clear, and some conclusions based on analysis of figures look declarative.

The curves in figures 2-6, plots B, are not explained, and it is impossible to follow their analysis.

Description of the model leaves unclear how it takes into account the influence of the ocean.

Estimates of the statistical confidence of trends should take into account the serial correlation as it is accepted in this field of analysis (see for instance Harris et al., Past changes in the vertical distribution of ozone – Part 3: Analysis and interpretation of trends, Atmos. Chem. Phys., 2015, vol 15, p. 9965-9982). This results generally in reduction in the number of degrees of freedom and, as a consequence, in worsening of statistical significance. Any conclusion about trends should be based only on statistically significant trend estimates. However, in the paper trends with very poor statistical significance are often discussed.

Specific comments (only part; it is laborious to list all)

L12. Abbreviation should be defined.

L26-29. This problem attracts attention for many years.

L28. Trend in reducing of content means that the rate of decrease in the content has trend.

L39-40.  More specifically, in the lower stratosphere.

L45-46. Is it actually was expected that the recovery should be linear?

L66. Define the abbreviation.

L89. Trends of ozone or trends of variability?

L96. Multidirectional periods?

L102. The mesopause is well above 90 km.

Figure 1C. What are units of aerosol surface?

L112. Fig, 1B shows sunspot number, not radiation fluxes. The source of sunspot data is not specified in the text.

L117. Does the abstract actually contain the spectral data of solar fluxes?

L146. What is the main trend?

L155. It is not the concentration.

L156. Selebrated?

L155-159. Absolutely unclear sentence.

L160-162. Where does that follow from?

L169-171. What is the base of this conclusion? Why the El Chichon eruption has but the Pinatubo eruption has not impact on ozone?

L172. What are global periodic fluctuations?

L172-173. What are twofold fluctuations? Do you mean the quasi-biennial oscillation?

L167-189. There are repetitions in this paragraph.

L194. What does negative variability mean?

L197 and further. Trend has units of DU per year.

L202-204. Repetition.

L208-211. Confusing statement. There is no ozone reduction in 2010-2020, according to fig. 3B.

L212 and further. Discharge gases?

L239-240. The trend is statistically insignificant.

L229-232. How can one discuss the “trend” for which the probability of the zero hypothesis is 90%?!

L235-236. Check the dates.

L246-247. What does “almost correct” mean?

L251-253. Clarify, why  it is overestimated.

L256-257. Why these peaks are attributed to the solar cycle? Its period is ~10 years, not 3-5 years.

L267. What is sustained variability?

L268. I do not see any significant positive trend in 1995-2000 in fig. 5.

L280-281. How can it be concluded from gig. 5?

L289. There is no ground to declare trend at this significance level.

L290-291. If so, why it is not the case in the Arctic? For more comprehensive comparison of model and observational results it is worth to average data over the same period of the year.

L292-294. I do not understand this sentence. How can it be deduced, whether the role grows or does not grow in 2010-2020?

L300-302. The trend is statistically insignificant in the Arctic, according to fig. 5.

L307-329. This is not related to the paper.

L336-337. Again, the evidence was not presented in the manuscript about the trend in the Antarctica in 2000-2018.

L342. Melamineware ozone – what is it?

L349-350. Unclear. In what sense it is optimistic?

Reviewer 2 Report

The paper is devoted to studying of ozone trends during the last four decades. This is very interesting and timely question for the scientific community and the public. Unfortunately, it is very difficult to read and understand the paper, in particular, because of the problems with English language. The paper needs extensive English editing and more clear explanations of the performed research.

Major comments:

The paper does not have description of the model. The Y axis of the Fig. 1A indicates only Chlorine concentrations while the plot shows two variables - Chlorine and Bromine concentrations.  Fig. 1B shows the dependence of SSN on time while the caption states that it is solar radiative flux.  The SBUV instrument does not measure any data in 2020 yet (line 134 ).  Fig. 2B, 3B, 4B, 5B and 6B have legend that is not explained neither in text nor in the caption to the figures.  All figures do not have titles of the X axis. The abbreviations are not always clear defined.  For example: ODS is Ozone-Depleting Substances in line 31; SBUV is Solar Backscatter Ultraviolet in line 70; OMI should be written by capital letters in line 82.
